# Obesity and Adipose Tissue Dysfunction: From Pediatrics to Adults

**DOI:** 10.3390/genes13101866

**Published:** 2022-10-15

**Authors:** Ana Menendez, Heather Wanczyk, Joanne Walker, Beiyan Zhou, Melissa Santos, Christine Finck

**Affiliations:** 1Connecticut Children’s Medical Center, Department of Pediatric Endocrinology, Hartford, CT 06106, USA; 2University of Connecticut Health Center, Department of Pediatrics, Farmington, CT 06030, USA; 3University of Connecticut Health Center, Department of Immunology, Farmington, CT 06030, USA; 4Connecticut Children’s Medical Center, Department of Pediatric Psychology and Director of the Obesity Center, Hartford, CT 06106, USA; 5Connecticut Children’s Medical Center, Department of Surgery and Pediatric Bariatric Surgery, Hartford, CT 06106, USA

**Keywords:** obesity, pediatric obesity, adipose immunity, immune system, pediatric obesity and inflammatory dysregulation

## Abstract

Obesity is a growing health problem that affects both children and adults. The increasing prevalence of childhood obesity is associated with comorbidities such as cardiovascular disease, type 2 diabetes and metabolic syndrome due to chronic low-grade inflammation present at early stages of the disease. In pediatric patients suffering from obesity, the role of epigenetics, the gut microbiome and intrauterine environment have emerged as causative factors Interestingly, pediatric obesity is strongly associated with low birth weight. Accelerated weight gain oftentimes occurs in these individuals during the post-natal period, which can lead to increased risk of adiposity and metabolic disease. The pathophysiology of obesity is complex and involves biological and physiological factors compounded by societal factors such as family and community. On a cellular level, adipocytes contained within adipose tissue become dysregulated and further contribute to development of comorbidities similar to those present in adults with obesity. This review provides an overview of the current understanding of adipose tissue immune, inflammatory and metabolic adaptation of the adipose tissue in obesity. Early cellular changes as well as the role of immune cells and inflammation on the progression of disease in pivotal pediatric clinical trials, adult studies and mouse models are emphasized. Understanding the initial molecular and cellular changes that occur during obesity can facilitate new and improved treatments aimed at early intervention and subsequent prevention of adulthood comorbidities.

## 1. Introduction

Obesity is a global health epidemic that affects both children and adults. Obese children are more likely to become obese adults [1] and it is estimated that almost half of the world’s adult population will be overweight or obese by 2030 [2]. The onset of obesity is occurring at younger ages in the last decade than previous generations [3]. Approximately 18.5% of youth in the United States meet the criteria for obesity (body mass index [BMI] ≥ 95th percentile for age and sex), while 8.5% of those 12 to 19 years of age are categorized as severely obese (BMI ≥ 120% of the 95th percentile), representing approximately 4.5 million children [4]. Most studies investigating obesity involve adults; however, it is essential for studies to focus on childhood/adolescent obesity to prevent its associated complications such as cardiovascular disease, type 2 diabetes mellitus, metabolic syndrome, and non-alcoholic fatty liver disease. Pediatric obesity has both near and long-term impacts as the physiological changes altered by obesity occur at crucial developmental stages. These comorbidities arise due to obesity-associated low-grade inflammation characterized by abnormal cytokine production and infiltration of macrophages into adipose tissue. Adipose tissue related inflammation leads to a wide variety of immune responses, involving early neutrophil participation followed by macrophage involvement and mast cell polarization [5]. These cellular adaptations lead to an altered metabolic profile early in life and a predisposition to premature mortality in adulthood. Studying the causes of obesity-associated inflammation in the pediatric population may identify opportunities to prevent progression to severe comorbidities such as hypertension, abnormal glucose metabolism and dyslipidemia. Susceptibility for development of childhood/adolescent obesity has also been linked to the in utero environment whereby maternal health plays a major role [6,7,8]. The effects of maternal obesity on inflammation in fetal development and early postnatal periods has been demonstrated in a variety of preclinical models where exposure to high fat maternal diets determine the incidence of offspring obesity, insulin resistance and fatty liver disease [9,10]. Similarly, other studies show that maternal obesity causes adipose tissue inflammation in young lean male mouse offspring [11,12]. It has also been demonstrated that maternal obesity can cause DNA methylation modifications at birth in several genes such as insulin growth factor binding protein-1 (*IGFBP-1*), which plays a role in the development of metabolic disease [13]. Studies show that low circulating levels of *IGFBP-1* lead to insulin resistance and potentially type 2 diabetes [14]. However, chronic obesity and its association with metabolic dysfunction needs to be better elucidated in the neonatal and pediatric population as comprehensive studies are lacking. Understanding early effects and risk factors associated with obesity will provide novel mechanistic insight into the disease process. In this review, we will highlight the early changes in the inflammatory-metabolic processes identified in the adipose tissue of obese populations in both clinical and pre-clinical studies and how these can lead to complications into adulthood.

## 2. Obesity in Early Childhood Years

Childhood obesity persists as a serious public health problem. Effective obesity prevention strategies and policies exist in children and adolescents; however, as reported in different clinical trials their positive effects are short-lived [15]. What is consistent in the data is that preventive interventions at early stages of life are found to be more beneficial than in adulthood. Therefore, addressing childhood obesity early may have some positive effects on long-term health [16]. It is important to understand the underlying genetics and individual behavior that varies among infants and young adults. It has been found that early metabolic processes including microvascular events begin in early childhood [17]. Normally, obesity is diagnosed in children 2 years and older based on their adjusted BMI [18]. However, because of limited studies in this age group, it is still unclear when to start screening for metabolic complications. Advances in metabolomics technology, epigenetics and single cell RNA sequencing are key drivers in gaining insight into the pathophysiology of obesity [19] since it allows a more comprehensive analysis of the adipose tissue. Adipose tissue plays a central role in energy homeostasis and production of adipokines, however, there are other factors that contribute to the development of comorbidities [20]. This review will focus on the physiology and pathophysiology of abnormalities within adipose tissue, highlighting multiple pre-clinical and clinical studies in both adults and children (Table 1) as well as preclinical animal models (Table 2).

## 3. Methods

A systematic search of the literature was performed on studies published between January 2000 and July 2022. The following keywords were used to search for relevant papers associated with childhood/adolescent obesity such as: childhood obesity or pediatric obesity; obesity-related inflammation; obesity-related endocrine system complications, obesity-related immune system, pathogenesis of low chronic inflammatory state within the adipose tissue, ectopic lipid accumulation; adipose tissue-associated inflammation; pro-inflammatory responses; dietary patterns; nutrients. The following electronic databases were searched: PubMed, EMBASE and Web of Science lastaccess on 30 of July 2022. The contributions were collected and analyzed, and the resulting data was assembled to provide a comprehensive review.

## 4. Composition of Adipose Tissue

Adipose tissue is a secretory organ capable of releasing endocrine hormones, cytokines, exosomes, miRNA, lipids, and peptide hormones that can act in a paracrine and autocrine manner. Adipose tissue can be categorized into three different cell types: white (WAT), brown (BAT), and beige (or brite) adipose tissue. White adipose tissue stores energy while BAT usually assists in maintenance of body temperature due to its high metabolic activity and high concentration of mitochondria. Additionally, it also contributes to energy expenditure. Brite adipose tissue is found scattered within the WAT and regulates the response to cold adaptation or other stimuli [34]. White adipose tissue is the most abundant in the body and is mainly distributed within the subcutaneous and visceral depots [35]. Visceral adipose tissue located within the intra-abdominal depots is associated with insulin resistance and metabolic disease, whereas accumulation of subcutaneous adipose tissue (located around the hips and flanks) has limited metabolic activity and may even be protective against metabolic syndrome [36].

Adipose tissue is composed of endothelial cells, mesenchymal stem cells, fibroblasts, pre-adipocytes, macrophages and several types of immune cells [37] located within the stromal vascular fraction (SVF) of the tissue (Figure 1A). Each cell population plays an important role in the maintenance and homeostasis of adipose tissue function. Comprehensive studies of adipose tissue cellular composition have been performed using traditional methods such as fluorescence activated cell sorting (FACS) and genetic lineage tracing, however these tools lack the ability to comprehensively identify and characterize dynamic interactions among stromal cells within the tissue microenvironment. Single cell RNA sequencing has emerged as a powerful tool to identify the cellular heterogeneity within the SVF. Single cell transcriptomic analyses of adipose tissue samples collected from visceral and subcutaneous depots have demonstrated that epithelial cells make up 8% of the population of cells, while 37% of the population are immune cells and 55% are adipocyte progenitor and stem cells [38]. Adipose-derived epithelial cells express genes such as fatty acid binding protein 4 (*FABP4*), glutathione peroxidase 3 (*GPX3*) and CD36 molecule, which are involved in fatty acid handling machinery [38]. Single cell RNA sequencing also revealed immune cells present across adipose depots with 40% having gene expression signatures of natural killer (NK) T cells, which are cells that bridge the innate and adaptive immune systems [38,39]. Additionally, it has been found that immune cells of adipose tissue express complement factor D (Adipsin) which is a known marker for adipocyte differentiation and is involved in triglyceride synthesis. High levels of adipsin correlate with an increased risk for development of diabetes [38,40]. Scavenger receptor CD36 expression within adipocytes and adipose tissue macrophages is also elevated in obese conditions, which can further contribute to metabolic dysregulation [41]. It has a critical role in large chain fatty acid (LCFA) uptake by adipocytes and provides a paracrine loop between adipocytes and macrophages [42]. When elevated, there is an increase in pro-inflammatory cytokines and adipocyte cell death.

The conformational changes of adipocytes and the cells contained within the SVF during obesity that contribute to development of comorbidities is an area of ongoing investigation. In general, the three main functions of the adipocyte within the white adipose tissue (subcutaneous or visceral depot) are lipid storage, secretory function and insulin sensitivity. Disruption of these essential functions during obesity has profound systemic effects [43]. Adipose depots become dysfunctional because of sustained lipid/nutrient overload leading to enlarged adipocytes that compromise global metabolic homeostasis.

## 5. Early Conformational Changes of the Adipocyte in Obesity

It is well known that obesity leads to a chronic inflammatory state and associated pathophysiological sequelae such as insulin resistance, altered lipid metabolism and ectopic lipid deposition (Figure 1B). Normally, early stages of adipocyte adaptation, whether located in subcutaneous or visceral depots, involve the release of controlled inflammatory signals that are required for proper adipose tissue remodeling and expansion including hyperplasia and hypertrophy. In an environment with excess caloric intake, adipose tissue expansion is an important systemic response to lipotoxic effects from fatty acids. Significant expansion of adipose tissue stimulates angiogenesis and extracellular matrix remodeling [44]. A study using a mouse model demonstrated that blocking the local pro-inflammatory response in adipocytes reduced adipose tissue growth [27]. Whether adipose tissue expansion in the development of obesity occurs primarily by hyperplasia or hypertrophy or due to adipose tissue dysfunction is a matter of debate [45]. When obesity and inflammation are sustained and the adipocyte diameter increases, a dysfunctional homeostatic process emerges, which is characterized by impaired secretion of adipokines, abnormal lipid storage, adipogenesis, increased fibrosis and insulin resistance [46]. During obesity, adipocytes can rapidly reach a threshold of oxygen limit during growth, therefore leading to hypoxic conditions.

### 5.1. Inflammatory Cascade Lead by Macrophages Propagates Severe Disease in Obese Conditions

The hypoxic environment created within adipocytes (excessive hyperplasia and/or hypertrophy) during an obese state has negative downstream effects that modulate the inflammatory response in both visceral and subcutaneous depots. One such response is the increased recruitment of monocytes and subsequent macrophage production. The primary immune response lead by macrophages involves pattern recognition receptors (PRRs) which include toll like receptors (TLRs). Toll like receptor 4 (TLR4) initiates saturated fatty acid mediated macrophage inflammation [47]. Obesity also increases the production of monocyte chemoattractant protein-1 (MCP-1) and leukotriene (LTB4), which in turn triggers the recruitment of monocytes that subsequently convert to mature adipose tissue macrophages (ATMs) [48]. Increased levels of MCP-1 is correlated with adiposity in both mice and humans. In adults, it was found that a BMI > 30 kg/m^2^ correlated with higher production of MCP-1 and C-reactive protein (CRP) levels [49]. In a cohort of obese Mexican American children, alterations in blood plasma cytokines/chemokine levels among healthy weight, overweight and obese children were found. Serum concentrations of MCP-1, interleukin-8 (IL-8) and tumor necrosis factor α (TNF-α) were higher in the obese vs healthy weight children [50]. Other findings in the adult population have included measurement of LBT4, which, when elevated, is implicated in macrophage recruitment and initiation of inflammation [51]. These studies demonstrate how macrophage function is significantly regulated by the surrounding microenvironment, which can further contribute to severe disease.

Adipose tissue macrophages (ATMs) are pivotal players in obesity associated inflammation and metabolic disease [52]. The two classic phenotypes of macrophages are M1 (pro-inflammatory) that releases cytokines in response to infection or stress and M2 (anti-inflammatory), which contribute to repair, remodeling and homeostasis [53,54]. Obesity studied in mice and adults confirm that the M1 macrophage phenotype and inflammatory pathway is predominant and that these macrophages are key players in propagation of associated inflammation and metabolic disease [55,56]. One response of activated pro-inflammatory macrophages during states of cellular hypoxia is to form crown like structures (CLS) around large dying adipocytes [57]. Macrophage infiltration into subcutaneous adipose tissue is enhanced in the obese pediatric population and CLS have been found in patients as young as 6–8 years of age [58]. In vitro studies demonstrate that stimulation of these macrophages with lipopolysaccharide (LPS) as well as interferon-γ (IFN-γ) increases macrophages (M1 phenotype), which promote a classical activation profile (M1) within the adipose tissue [59]. However, under the influence of the TH2 cytokine interleukin (IL)-4 and IL-13, macrophages assume an alternative activation state (M2a) and produce immunosuppressive factors, such as IL-10, which is an anti-inflammatory cytokine that also promotes insulin sensitivity [60]. This alternative/anti-inflammatory activation pathway is mediated by IL-4, which is expressed in high amounts in lean adipose tissue. IL-4 induces the expression of peroxisome proliferator activated receptor (PPAR-) gamma (γ) and delta (δ), which are required for maintenance of this anti-inflammatory pathway [28].

Additionally, macrophages are capable of secreting chemotactic molecules such as tumor necrosis factor α (TNF-α) [61]. TNF-α has been described as a potent mediator that recruits cytokines such as IL-6, IL-1β, MCP-1, and macrophage inhibitory factor (MIF) [60]. The secretion of TNF-α leads to insulin resistance following phosphorylation of serine residues of the insulin receptor substrate-1. this prevents the receptor substrate from binding to the insulin receptor, leading to inhibition of insulin activity [62]. Increased IL-6 production leads to elevated liver CRP resulting in a prolonged inflammatory state. CRP along with complement activation are acute phase reactants that can lead to cytokine production that increases the chronic inflammatory state [62]. Increased levels of IL-6, CRP and TNF-α are present in children with obesity, which increases atherosclerotic risk factors [63]. Even in age- and sex-adjusted models, IL-6 is associated with elevated insulin, insulin resistance, BMI, and waist circumference [64]. In subcutaneous adipose tissue collected from adults with obesity, it was found that TNF-α expression is also increased [65,66]. These inflammatory mediators also stimulate de novo hepatic lipogenesis, contributing to hepatic steatosis and elevated serum lipid levels [67]. In a mouse model of obesity, administration of intravenous TNF-α or IL-1β increased the activity of acetyl-CoA carboxylase, an enzyme that plays a role in regulation of lipid synthesis [67], and also resulted in hyperlipidemia.

Additionally, TNF-α, IL-6 and MCP-1 have been observed in the serum of individuals with diabetes and elevated production of TNF-α leads to increased serum glucose concentrations [68,69]. In a rodent model of obesity, normalization of TNF-α decreased insulin resistance [30]. In addition, a targeted mutation of the gene encoding TNF-α resulted in improved insulin sensitivity with lower levels of free fatty acids and normal expression of insulin receptor signaling in fat tissues and muscle [31]. Adults with obesity have been found to have elevated serum levels of TNF-α [69], and this positively correlates with the development of type 2 diabetes. Adipokines such as leptin, IL-6, retinol binding protein-4 (RBP-4), and angiopoietin like protein 2 (ANGPTL2) play a crucial role in the inflammatory immune reaction and are elevated in pediatric obese population. Not only do they cause metabolic disease, but they have also been linked with asthma, childhood multiple sclerosis and rheumatologic diseases [70,71,72]. Further investigation into the factors secreted by macrophages during obesity may aid in the development of novel therapies/interventions to prevent severe disease progression.

### 5.2. Other Innate Immune Cells in Obesity

Recent studies report that immune cells such as neutrophils, mast cells, and eosinophils, infiltrate adipose tissue and lead to chronic inflammation [29]. In the pediatric and adult population, adipose tissue-related inflammation leads to increased circulating neutrophils and monocytes suggesting that obesity alters leukocyte production and/or turnover resulting in abnormal insulin signaling [73]. In obese mice and adult human subjects, it was found that after feeding a high fat diet, neutrophils are recruited to adipose tissue. They normally circulate in the blood in a resting state, however in times of tissue damage they are recruited by IL-8, complement 5a (C5a) and LTB4. Once activated, they secrete pro-inflammatory cytokines and serine proteases [74]. Neutrophil elastase (a serine protease) can alter insulin signaling as demonstrated in murine studies [75]. Additionally, mast cells are rapidly recruited in white adipose tissue obtained from obese adults and mice fed a high fat diet. Mast cells secrete granules that are rich in histamine and cytokines such as TNF-α and IL-1β, which further exacerbate inflammatory conditions [76]. Tryptase levels are used as an indicator of increased mast cells circulating in the blood. A study conducted in obese, overweight and lean individuals from a pediatric population (ages 8–18 years old), found no statistical significance between higher BMI and levels of tryptase, which is contrary to what has been reported previously [77]. However, additional studies are needed to fully establish and characterize the importance of mast cells in obesity progression within the pediatric population.

Eosinophils serve as a negative regulator of adipose tissue inflammation. They are present in the SVF of lean adipose tissue and produce IL-4, a driver of the M2 anti-inflammatory pathway. In obese mice, the concentration of eosinophils is diminished, resulting in the loss of this protective pathway [28]. Furthermore, blood derived NK cells response appears to be impaired in obesity due to a decrease in their cytotoxic capabilities [78,79,80]. In a cohort of obese children (ages 6–16), without metabolic abnormalities, it was found that there were less numbers of NK cells compared to normal controls. Additionally, these cells had defective production of granzyme B and perforin, molecules both responsible for tumor lysis function by NK cells [81]. A study reported that patients who had bariatric surgery between the ages of 23–58 years old with a BMI > 40 kg/m^2^ demonstrated normalized NK cell functionality within 6 months post-surgery [82]. Understanding the role of different immune cells in obesity can elucidate the mechanisms that contribute to developing serious comorbidities such as increased risk of cancer and may serve as an improved means to determine treatment response.

### 5.3. The Role of Adipose Tissue Adaptive Immune Cells in Inflammatory Progression during Obesity

In addition to innate immune cells, T cell and B cell activation during the adaptive immune response also plays a large role in the regulation of adipose tissue inflammation. Several studies have demonstrated that these cells reside within the adipose tissue and regulate inflammation [83,84]. CD4+ T cells are divided into regulatory T (Treg) cells and T helper (Th) cells. The cytokines produced by the subset T helper 1 cells, are responsible for pro-inflammatory cytokine release of INF-γ, IL-2 and TNF-α, which are increased in obesity [84]. In a study comparing mice fed a high fat diet vs. a normal diet, it was found that even before the accumulation of macrophages, T cells infiltrated adipose tissue and perpetuated the production of inflammatory cytokines. Similarly, M2 like macrophages, CD3+ T cells, CD4+, and Treg cells are important negative regulators of visceral adipose tissue inflammation. Treg cells have been found enriched within the abdominal fat of normal mice [29]. However, their numbers are significantly decreased in the insulin resistant models of obesity [29]. Loss-of-function and gain-of-function experiments revealed that these Treg cells influence the inflammatory state of adipose tissue and, thus, insulin resistance [85].

Additionally, studies in preclinical models of obesity demonstrate that B cells play a critical role in adipose tissue inflammation since they infiltrate visceral adipose tissue in response to high fat diets. Mice fed a high fat diet also demonstrated early accumulation of immune cells and increased T cell numbers, which correlated with the incidence of insulin resistance and macrophage accumulation in adipose tissue [86]. In mice it is known that B cells can infiltrate visceral adipose tissue in response to a high fat diet and activate pro-inflammatory macrophages and T cells, which lead to disease. In the same study, it was found that a pathogenic variant of immunoglobulin G antibodies (IgG) can cause increased levels of the TNF-α and M1 activation pathway [87]. Several mouse models suggest that obesity impairs antibody production and accelerates insulin resistance [32]. In an obese mouse study, B cells produced an elevated inflammatory cytokine profile compared to lean mice [88]. B cells can also be activated either by altered lipolysis, as seen in obesity, and through local or systemic inflammatory cytokines [89]. In adults, there is a positive correlation between lymphocytes in serum samples from subjects with an elevated BMI and inflammatory diseases such as asthma [90]. Human peripheral blood samples collected from type 2 diabetic patients compared to control patients (non-diabetic) demonstrated an altered B cell activation leading to an increased cytokine production of IL-8 and promoting chronic inflammation [91]. B cells can also promote an alteration of the adipocytes by hypertrophy [92]. Therefore, B cell cytokine secretion and/or antibody production across obesity models needs to be further investigated since this appears to play a contributory role.

### 5.4. Metabolic Adaptations in Obese Adipose Tissue

Adipose tissue is a highly dynamic organ and is capable of responding to environmental inputs with hypertrophy (for lipid storage) or hyperplasia. Alterations in the tissue occur within days of an obesogenic diet [93]. Maladaptation in adipose tissue can lead to metabolic disorders such as insulin resistance, dyslipidemias, hepatic steatosis and type 2 diabetes. Chronic inflammation in white adipose tissue leads to adipocyte malfunction by dysregulation of mitochondrial biogenesis and oxidative phosphorylation, with excess accumulation of extracellular matrix, altered adipokines and insulin resistance [94]. Mitochondria dysfunction was tested in an experimental study of mice deficient in a nuclear encoded transcription factor A (a gene-master regulator of the mitochondrial biogenesis, located in the mitochondria of the adipocyte). Mitochondrial dysfunction led to adipocyte death, adipose tissue inflammation and systemic insulin resistance [95]. In obese patients or non-diabetic human subjects, elevated chronic glucose levels caused hyperinsulinemia, which is also linked to a high fat diet. Hyperinsulinemia in both rats [96] and humans [97] enhances activation of inflammatory pathways, which in turn can impair insulin responsiveness in target tissues [98].

The decreased capacity of adipocytes to continue to store and retain triglycerides in obesity, causes ectopic fat accumulation in liver and muscle leading to insulin resistance, which is considered the key mediator in cardio metabolic disease [99]. Emerging evidence in clinical studies suggests that extracellular matrix proteins such as collagen, fibronectin and elastin promote adipocyte remodeling and fibrosis. Elevated levels of proteins are associated with pro-inflammatory immune cells such as M1 macrophages [100,101]. In adult studies, it has been shown that increased fibrosis within the adipocytes leads to a lower response to interventions such as weight loss and bariatric surgery [102]. Human and rodent studies differ in their metabolic response to fibrosis within adipose tissue, therefore caution needs to be taken when comparing results between the two different species. In rodent studies, visceral adipose tissue is more prominent than in humans and is responsible for the dysregulated metabolism. In an obese mouse model, it was found that circulating TNF- α and transforming growth factor β (TGF-β) were key regulators of adipocyte tissue fibrosis [103]. Visceral and subcutaneous adipose tissue collected from obese humans during bariatric surgery demonstrated decreased fibrosis and pre-adipocyte frequency along with increased adipocyte hypertrophy. This supports the idea that decreased adipose tissue fibrosis in metabolic disease is associated with a tissue architecture that permits adipocyte hypertrophy and limits pre-adipocyte hyperplasia, leading to larger, metabolically impaired adipocytes [100]. Data pertaining to the degree of adipocyte fibrosis experienced in the pediatric population is lacking.

The sympathetic nervous system is also implicated in the mobilization of fatty acids from white adipose tissue. Ganglionic sympathetic nerves innervate adipocytes [104]. Norepinephrine can act on the adipocyte membrane using βeta-adrenergic receptors (β-ARs), and this results in the release of free fatty acids into the circulation from stored triglycerides [105]. Elderly adults with increased visceral adiposity have a decrease in the negative regulators of adipocyte lipolysis that catabolize the local extracellular norepinephrine. These regulators, called sympathetic neuron associated macrophages (SAMs), have specialized transporters with monoamine oxidase [106]. These SAMs affect mobilization of free fatty acids and its usage for energy and metabolism. Genetically engineered mouse models with a defect in the β-ARs demonstrate that high caloric intake results in a reduced metabolic rate and predisposition to obesity [107].

Adipocyte cell size has also been found to have an impact on glucose and lipid metabolism. In morbidly obese women, mean visceral and subcutaneous fat cell sizes were found to be related to in vivo markers of inflammation, glucose metabolism and lipid metabolism. These associations were independent of age, BMI or body fat distribution [108].

## 6. Impact of Pediatric Obesity and Subsequent Adipose Tissue Dysfunction on Long-Term Health

In childhood obesity, cells within the SVF including macrophages, NK cells, B cells and T cells are dysregulated [109]. Long-term childhood obesity leads to abnormal immune function of adipose tissue that result in serious comorbidities. Studies in mice suggest that early intake of a high fat diet causes adipocytes to have uncoupled respiration, leading to increased oxygen consumption, a state of relative adipocyte hypoxia and inflammation [33]. Intracellular inflammatory cascades such as c-Jun N-terminal kinase (JNK) and nuclear factor kappa β (NF-κβ) are activated when conformational changes within the adipocyte occur due to increased free fatty acids from a high fat diet [110]. In response to these pathways, there is activation and recruitment of macrophages, B cells, T cells and other inflammatory cells within adipose tissue. This suggests that adolescents fed a predominately high fat diet, which is common in the United States, can lead to inflammation through activation of these pathways. Early studies using a gene expression omnibus database demonstrated subcutaneous adipose tissue from seven obese children had an upregulation of matrix metalloproteinase-9 (*MMP9*), when compared with eight lean children. *MMP9* is a family of zinc-dependent endopeptidases that interact with components of basement membranes and extracellular matrix This upregulation led to hypertrophy and hyperplasia of adipocytes [21]. Adipose tissue collected from obese children showed CLS formation around dead adipocytes in visceral adipose tissue similar to the changes seen in adults [22].

Children suffering from obesity (ages 6–17 years of age), have lower levels of NK cells in peripheral blood when compared to lean children at the same age. This affects the ability of their immune system to destroy infected cells or initiate inflammatory cascades against abnormal cells as previously described [81]. In a mouse model, the lack of NK cells within the adipose tissue was related to adipose tissue weight gain, larger adipocytes, fatty livers, and insulin resistance [23]. In vitro studies demonstrated that the stimulation of NK cells by cytokines such as IL-5 and IL-2 differed when isolated from lean samples compared to obese samples. NK cells from obese samples did not proliferate and showed no expansion compared to NK cells from lean samples that had continuous proliferation [81]. Additionally, the adaptive immune system is also implicated in initiating insulin resistance by increasing inflammation in obese children. In a cohort of pediatric pre-pubertal obese patients, IL-10 was found to be decreased in peripheral blood samples leading to inflammation [24]. Children with obesity and metabolic syndrome exhibit recruitment of CD8 T cells, which compounds inflammation.

Interestingly, exercise is an effective mitigation strategy in preventing metabolic disease in children. Increased physical activity leads to improvement in adipocyte expansion in the visceral and subcutaneous tissue [111]. Animal models demonstrated that as little as 4 weeks of aerobic exercise prevents conformational changes in adipose tissue [112]. In an early systematic review, physical activity with aerobic exercise inhibits adipose tissue paracrine and endocrine inflammatory responses [113]. This may result in children being less susceptible to infections. As we know, children and adolescents with obesity have increased susceptibility to COVID-19 infection [114] and influenza infection through the effect of hyperlipidemia and hyperglycemia on the T cell response [115]. The idea of introducing exercise to improve metabolic and immune response is intriguing and requires further study in children [115]. Caloric restriction may also improve health, as demonstrated in a mouse model where it led to immune cell redistribution and improved fatty acid oxidation [116]. Therefore, in childhood obesity, strategies to reverse the inflammation and immune dysregulation are desperately needed. Despite data presented in this review, there is a need for further investigation into the pathophysiology of pediatric obesity induced inflammation and conformational adipocyte changes to help prevent the development of lifelong comorbidities.

## 7. Blood Biomarkers as a Reflection of Adipose Tissue Dysfunction in Pediatric Patients

Peripheral blood cells (PBC) can reflect the inflammatory profiles of internal tissues. This is important as adipose tissue collection from pediatric populations is challenging without surgical intervention. A cross sectional analysis of inflammatory markers such as CRP, neutrophil count and ferritin/transferrin ratio in children ages 1 to 17 years old was completed. CRP levels of >1.0 mg/L were evident among children with obesity as early as 3 years old. In children 6 to 8 years of age, an abnormal neutrophil count was noted among children with severe obesity and an increased prevalence of abnormal ferritin/transferrin ratio began at 9 to 11 years of age, demonstrating how early obesity increases inflammation [26]. Leptin, a protein produced by fatty tissue that regulates fat storage, was elevated in obesity, altered immune function through T cell receptors, increased interferon-γ (IFN-γ) secretion and decreased IL-4 secretion [117]. Interestingly, leptin levels were increased in the obese pediatric populations [117] and seen in blood as early as 12 years of age [118]. In patients who had weight loss by physical activity including aerobic exercise, M1 macrophages along with inflammatory cytokines are decreased [113] proving how early interventions can diminish this inflammatory cascade.

## 8. Conclusions

Obesity is associated with a chronic inflammatory state that is present in the pediatric population. The effects of obesity in pediatric patients can be long lasting, even with a normalized BMI by adulthood [119]. Adipose tissue contains a vascular stromal fraction that when dysregulated (Figure 1A), can cause alterations in normal homeostasis. When adipocyte conformational changes occur, immune dysregulation leads to metabolic effects such as increased lipolysis and inflammation, and decreased insulin sensitivity (Figure 1B). Studies have shown that an upregulation in inflammation occurs early in childhood obesity. Increasing knowledge on early immune and inflammatory changes seen in obesity can lead to a better understanding on possible mechanisms to restore this altered homeostasis and prevent the development of lifelong comorbidities. It is important to mention that bariatric surgery results in an early (<3 months) reduction of total adipose tissue macrophages found in subcutaneous adipose tissue. In addition, there is a remission of the CLS [120] 6 months post-surgery, and a decreasing inflammatory profile. Caloric restriction and aerobic exercise can also improve adipose immune cell function in children with obesity, leading to major redistribution of circulating leukocytes. Studies in the pediatric population will allow better insight into the early processes occurring in obesity. In addition, children usually present with earlier stages of disease. Therefore, studying the underlying mechanisms would be more accurate since there are less preexisting comorbidities. Further work in the robust characterization of the inflammatory markers and adipose tissue components in the pediatric population are needed.

## Figures and Tables

**Figure 1 genes-13-01866-f001:**
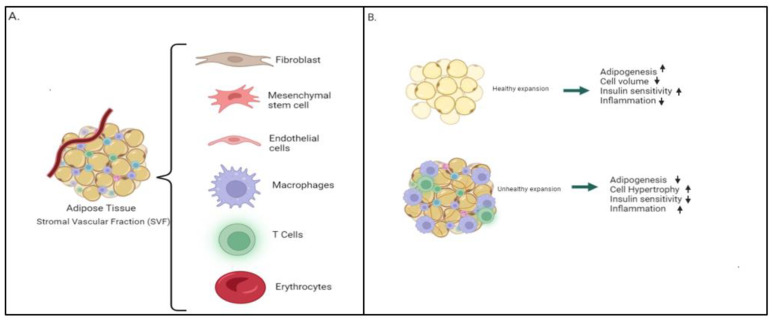
(**A**) The adipose tissue is primarily comprised of adipocytes which are surrounded by various vascular stromal cells such as depicted in the figure. 1/3 is formed by vascular stromal cells including fibroblast, mesenchymal stem cells, endothelial cells, macrophages, T cells. 2/3 of the fraction will be composed of adipocytes. (**B**) Adipose tissue dysfunction is a determinant of obesity causing metabolic complications. Lipotoxicity leads to obesity through ectopic deposition of lipids, Nonalcoholic fatty liver disease (NAFLD), insulin resistance, T2 diabetes and coronary heart disease.

**Table 1 genes-13-01866-t001:** Summary of Childhood Obesity published manuscripts.

Reference Paper	Title	Study Design	Major Finding
Li L 2017, [21]	Identification of key genes and pathways associated with obesity in children	Subcutaneous adipose tissue collected from pediatric obese and lean patients	Matrix metallopeptidase 9 (MMP9) is upregulated in immune system process and has a significant role in child obesity
Jaitlin DA 2019, [22]	Lipid associated macrophages control metabolic homeostasis in a Trem2-dependent manner	Mapping of adipose tissue immune populations in both mice and humans	Identification of associated macrophages subsets with markers and functional pathways associated with adipose tissue and obesity
O’shea D 2019, [23]	Dysregulation of natural killers in obesity	Blood samples collected from obese patients	NK cells have a role in homeostasis and if not functional contribute to pathophysiology of inflammation
Bassols J 2014, [24]	Increased serum Immunoglobulin G (IgG) and Immunoglobulin A (IgA) in overweight children relate to a less favorable metabolic phenotype	Blood from pediatric obese population	Increased circulating IgG and IgA in overweight children are associated with metabolic phenotype
Yang Dh 2021, [25]	Effector memory CD8^+^ and CD4^+^ T cell immunity associated with metabolic syndrome in obese children	Blood from pediatric obese population	In obese children, the changes in CD8^+^ T cells might be related to the morbidity of obesity
Skinner Ac 2010, [26]	Multiple markers of inflammation and weight status: cross-sectional analyses throughout childhood.	Blood samples from pediatric population	Inflammatory markers are strongly and positively associated with increasing weight status in children
Sasaki A 2022, [13]	DNA methylation profiles in the blood of newborn term infants born to mothers with obesity	Newborn blood sample collected by heel stick	DNA methylation modifications were associated with maternal obesity

**Table 2 genes-13-01866-t002:** Summary of Obese animal models publish manuscripts.

Reference Paper	Title	Study Design	Major Finding
Wernstedt 2014, [27]	Adipocyte inflammation is essential for healthy adipose tissue expansion and remodeling	Three mouse models with adipose tissue	High fat diet expands the adipocyte, increases hepatic steatosis and metabolic dysfunction.
Wu 2011, [28]	Eosinophils sustain adipose alternatively activated macrophages associated with glucose homeostasis	White adipose tissue analyzed from mice model	Mice on high fat diet develop increased body fat, impaired glucose tolerance and insulin resistance in the absence of eosinophils.
Feuerer 2009, [29]	Lean, but not obese, fat is enriched for a unique population of regulatory T cells that affect metabolic parameters	Mice model highly enriched in abdominal fat	Loss of function and gain of function experiments revealed that T (reg) cells influence inflammatory state of adipose tissue and thus insulin resistance.
Hotamisligil 1993, [30]	Adipose expression of tumor necrosis factor-α direct role in obesity-linked insulin resistance	Mouse models with induced high fat diet	TNF-α RNA expression was observed from four different rodent models
Uysal 1997, [31]	Protection from obesity-induced insulin resistance in mice lacking TNF-α function	Obese mice with a targeted null mutation in the gene encoding TNF-α	The absence of TNF-α resulted in significantly improved insulin sensitivity in diet induced mice obesity.
Arai S 2013, [32]	Obesity-associated autoantibody production requires AIM to retain the immunoglobulin M immune complex on follicular dendritic cells	Obese mouse model AIM (apoptosis inhibitor macrophages) deficient	IgM-AIM association contributes to autoantibody production under obese conditions
LEE 2014, [33]	Increased adipocyte O2 consumption triggers HIF-1alpha, causing inflammation and insulin resistance in obesity	Obese mouse model	early in the course of high-fat diet (HFD) feeding and obesity, adipocyte respiration becomes uncoupled, leading to increased oxygen consumption and a state of relative adipocyte hypoxia

## Data Availability

Not applicable.

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
