# Peer review of "Obesity and Adipose Tissue Dysfunction: From Pediatrics to Adults"

_genes, 2022, doi:10.3390/genes13101866_

Round 1
Reviewer 1 Report
This manuscript entitled "Pediatric Obesity and Adipose Tissue Dysfunction" by Ana Menendez et al, according to the abstract and title, focuses on immune and metabolic changes in adipose tissue in the context of obesity with a focus on the pediatric population. The chosen topic is very interesting and important at the present time. However, in fact this manuscript only focuses on the immune response (and inflammation) in adipose tissue of obese adults or rodents fed a high-fat diet, and metabolic changes are only minority mentioned. Only two sections at the end of the manuscript are devoted to childhood obesity and adipose tissue dysfunction.
Chapter 3. is focused on composition of adipose tissue, where are mentioned 2 types of adipose tissue, brown and white, but brite adipocytes are not included. The following sentence describes visceral and subcutaneous adipose tissue and there is no note that these are types of white adipose tissue. Unfortunately, in later chapters, where immune changes and inflammation are desribed, we do not find whether these changes involve both tissue or whether differ from each other. The second part of this chapter (Immune cells were ….) describes the composition of immune cells in adipose tissue. However, different clusters are described that have their origins in the cited publication and this section is unclear.
In chapters 4-6 authors desribe immune changes during obesety including child obesity in chapters 5,6. However, the information described is not sufficiently well arranged for the reader.
It is not clear from the content of the manuscript whether the author's aim is: to desribe immune response in adipose tissue in obesity or to describe the difference between childhood and adult obesity or its impact in adulthood. It is not clear from the content of the manuscript whether the author's aim is to describe the difference between childhood and adult obesity or its impact in adulthood. In any case, the manuscript should be revised according to the aim.
Minor comments:
1) Please check names and afiliations
2) Abbreviations in the manuscript are unconsistent, gene abbreviations are often without prior introduction – names of genes missing. Some abbreviations are introduced but they are not used in the following text (or used only sometimes)
3) there is no reference to figures and table in the text
4) missing marks a, b in the figure
5) for no reason, capital letters for gene names and other words in the middle of sentences
6) Figure desription: correct please 2.3 into 2/3
Reviewer 2 Report
Dear Authors
This is an interesting and well-written article.
Comments
· All abbreviations used in table 1 should be defined in the table note.
· Please add a table summarizing the main findings of obesity-related animal studies.
· Please check Figure 1.
· Please cite Figure 1 in the Conclusions section and add some explanations.
· Please add a graphical abstract (see Instructions for Authors).
· Please correct writing errors throughout the manuscript.
Round 2
Reviewer 1 Report
The manuscript was revised and extensively rewritten. The most comments were included and edited.
Only minor errors should be corrected:
Page 2, lines 78 – 77, sentence “The use of male mice was instrumental in controlling for the metabolic influence of female sex hormones” should be excluded from manuscript. The statement is not accurate
Page 5, line 189, edit word combinations “at the adipocytes”
Page 6, line 222, I would leave out “Macrophage” and follow up with a paragraph “Adipose tissue…”
Line 247-248 thanks to the introduction of the abbreviation for PPAR gamma, you can leave out “peroxisome proliferator activated receptor” after “and ….”,
Page 6 after introduction of abbreviation for interleukin (IL), you can use father only IL and number
Page 7, line 264 “de novo” in italic
Page 7, line 271 “Cytokines:” , I would leave out and follow up with a paragraph “Additionally, …”
Page 8, line 313, use only SVF (established line139)
Line 318. Use only NK (established line155)
Page 9, line 348 Leave out (VAT)
Page 9-10, line 377-379, please improve the sentence “Patients with type 2 diabetes …..”. It is not clear, if the B cells are only in these patients or the amount is increase and where?
Page 11, line 442 use only SVF (established line139)
Line 443. Use only NK (established line155)
Page 12, line 467 use only CLS (established line 233)
Line 481. Use only NK (established line155)
Page 13, line 518 use only CRP (established)
line 539 use only SVF (established line139)
Page 14, line 550 use only CLS (established line 233)
Reviewer 2 Report
Please remove the duplicate figure from the first row of Table 2.
